# Liver Involvement in Patients with Systemic Sclerosis: Role of Transient Elastography in the Assessment of Hepatic Fibrosis and Steatosis

**DOI:** 10.3390/diagnostics13101766

**Published:** 2023-05-17

**Authors:** Giovanna Cuomo, Carlo Iandoli, Raffaele Galiero, Alfredo Caturano, Claudio Di Vico, Danilo Perretta, Pier Vincenzo Adamo, Roberta Ferrara, Luca Rinaldi, Ciro Romano, Ferdinando Carlo Sasso

**Affiliations:** 1Department of Precision Medicine, “Luigi Vanvitelli” University of Campania, 80138 Naples, Italy; 2Department of Advanced Medical and Surgical Sciences, “Luigi Vanvitelli” University of Campania, 80138 Naples, Italyciro.romano@unicampania.it (C.R.);

**Keywords:** systemic sclerosis, FibroScan, liver fibrosis, liver steatosis

## Abstract

*Background* Systemic sclerosis (SSc) is a rare, multisystemic disorder of connective tissue characterized by widespread inflammation, vascular abnormalities, and both skin and visceral organ fibrosis. Tissue fibrosis is the final phase of a complex biological process of immune activation and vascular damage. *Objectives* The aim of the study was to assess hepatic fibrosis and steatosis in SSc patients by transient elastography (TE). *Methods* Fifty-nine SSc patients fulfilling the 2013 ACR/EULAR classification criteria were recruited. Clinical and laboratory findings, modified Rodnan skin score (mRSS), activity index, videocapillaroscopy, echocardiography, and lung function data were analyzed. Liver stiffness (LS) was measured by transient elastography (TE), with 7 kPa used as the cut-off value for significant liver fibrosis. In addition, hepatic steatosis was evaluated by means of controlled attenuation parameter (CAP) findings. Specifically, CAP values ≥ 238 ≤ 259 dB/m were considered consistent with mild steatosis (S1), values ≥ 260 ≤ 290 dB/m were compatible with moderate steatosis (S2), and values ≥ 291 dB/m were indicative of severe steatosis (S3). *Results* The median age of patients was 51 years, with a median disease duration of 6 years. The median LS was 4.5 (2.9–8.3) kPa; 69.5% of patients had no evidence of fibrosis (F0); 27.1% displayed LS values between 5.2 and 7 kPa; and only 3.4% of patients had LS values > 7 kPa (F3). The median CAP value for liver steatosis was 223 dB/m (IQR: 164–343). Overall, 66.1% of patients did not show evidence of steatosis (CAP values < 238 dB/m); 15.2% showed values consistent with mild (S1) steatosis (CAP value ≥ 238 ≤ 259 dB/m); 13.5% had moderate (S2) steatosis (CAP value ≥ 260 ≤ 290 dB/m); and 5.1% were deemed to have severe steatosis (S3) due to CAP values ≥ 291 dB/m. *Conclusions* Although systemic sclerosis is associated with fibrosis of the skin and several organs, only 3.4% of our patient population showed evidence of marked liver fibrosis, which is the same prevalence as that expected in the general population. Therefore, fibrosis of the liver did not appear to be a significant concern in SSc patients, albeit moderate fibrosis could still be detected in a significant proportion of subjects. A prolonged follow-up may clarify whether liver fibrosis in SSc patients may still progress. Likewise, the prevalence of significant steatosis was low (5.1%) and depended on the same variables associated with fatty liver disease in the general population. TE was shown to be an easy and valuable method for detection and screening of hepatic fibrosis in SSc patients with no additional risk factors for liver disease and may be useful to assess the potential progression of liver fibrosis over time.

## 1. Introduction

Systemic sclerosis (SSc) is a chronic multisystem connective tissue disease of unknown etiology and intricate pathogenesis [1]. The pathophysiology of SSc is indeed still under investigation, whereby chronic inflammation, deposition of collagen, and fibrosis of the affected tissues may lead to thickening of the skin and possible involvement of internal organs, mainly the lungs, the gastrointestinal tract, the heart, and the kidneys. Briefly, vascular damage (proliferative and obliterative vasculopathy) in genetically susceptible individuals, triggered by environmental factors, prompts activation of endothelial cells, recruitment of innate and adaptive immune cells, and progressive fibrosis of internal organs [1,2]. Fibrotic changes of internal organs, such as the lungs, heart, kidneys, and gastrointestinal tract, characterize the clinical course of both limited (lcSSc) and diffuse cutaneous SSc (dcSSc), potentially leading to organ dysfunction [3]. However, liver involvement by fibrosis is less delineated. Indeed, data are scarce about the real prevalence of disease-associated fibrotic changes in the liver of SSc patients, which appear to be around 13% according to the little literature available [4,5]. Notably, liver involvement in SSc, although uncommon, is a well-known complication for rheumatologists caring for these patients. However, when this occurs, it is mainly due to autoimmune comorbidities such as primary biliary cholangitis (PBC), autoimmune hepatitis, or overlap syndromes [6,7]. Of these, primary biliary cholangitis is the most common cause of liver involvement in SSc patients [7]. This is not surprising, since both conditions are deemed to be autoimmune diseases. Indeed, several observations point to a shared autoimmune pathogenesis between SSc and PBC, among which: (1) the detection of SSc characteristic anti-centromere antibodies, which are associated with the limited SSc subset, in 9–30% of PBC patients with no apparent concurrent connective tissue disease; (2) the detection of the characteristic PBC anti-mitochondrial antibodies (AMA) in 25% of SSc with no apparent hepatobiliary involvement; (3) the relatively frequent association of SSc and PBC with other organ-specific and/or systemic autoimmune conditions, among which Hashimoto’s thyroiditis, Graves’ disease, Sjögren’s syndrome, and rheumatoid arthritis are the most frequently observed [7]. Thus, it is not surprising that SSc patients are first evaluated for an overlapping autoimmune hepatobiliary condition in the case of abnormal results of liver biochemistry tests rather than for fibrosis also involving the liver along with the other organs and tissues characteristically damaged by the disease. A deeper knowledge is therefore required to assist clinicians in decision-making, particularly to establish the need for liver assessment when a diagnosis of SSc is made and for liver fibrosis monitoring during the clinical course of the disease. Transient elastography (TE) is a relatively new diagnostic method able to analyze the mechanical characteristics of the investigated tissues, thereby yielding non-invasive images of their density (elasticity). This technology has evolved rapidly in recent years [8,9]. Briefly, the device is equipped with a probe capable of emitting ultrasound, which produces an elastic wave passing through the liver parenchyma. Liver stiffness (LS) is then deduced by the software on board the machine following analysis of the data related to the crossing speed of the elastic wave in the liver. The final output is a numerical value for each measurement, expressed in kilopascal (kPa) [8,9]. Thus, TE may allow non-invasive measurement of liver fibrosis, overcoming the need for a liver biopsy. Moreover, TE has also evolved to provide an estimate of fat infiltration in the liver and is currently widely used by hepatologists to non-invasively grade hepatic steatosis.

The aim of this study was, therefore, to evaluate the prevalence of hepatic fibrosis and fatty liver in SSc patients by means of TE. Secondary endpoints were the identification of predictors of liver steatosis or fibrosis and their analysis.

## 2. Materials and Methods

### 2.1. Study Population

A cross-sectional study was carried out on 59 patients at the Rheumatology Clinic of the “Luigi Vanvitelli” University of Campania, Naples, Italy. From January 2022 to June 2022, the clinical records of all patients aged ≥ 18 years were retrieved from the clinic’s database. SSc is considered a rare disease in Italy, and its overall nationwide annual prevalence is estimated to be 306.1 (95% CI 301.1–311.2) per million [10]. This indeed explains why only 59 patients could be recruited at a single site. Both limited and diffuse SSc patients [1] were included in this study. The SSc diagnosis was based on the American College of Rheumatology (ACR)/European League Against Rheumatism (EULAR) 2013 classification criteria [11]. Briefly, the following items were considered: skin thickening of the fingers of both hands extending proximal to the metacarpophalangeal joints (sufficient stand-alone criterion); skin thickening of the fingers (puffy fingers or sclerodactyly of the fingers distal to the metacarpophalangeal joints but proximal to the proximal interphalangeal joints); fingertip lesions (digital tip ulcers, fingertip pitting scars); telangiectasia; abnormal nailfold capillaries; pulmonary arterial hypertension and/or interstitial lung disease; Raynaud’s phenomenon; SSc-related autoantibodies (namely, anti-centromere; anti-topoisomerase I [anti-Scl-70]; and anti-RNA polymerase III). Each of the above criteria was weighted differently, as per ACR/EULAR guidelines: for patients to be classified as having definite SSc, a total score of ≥9 had to be reached [11]. Scleroderma-mimicking conditions, including nephrogenic sclerosing fibrosis, generalized morphea, eosinophilic fasciitis, scleredema diabeticorum, scleromyxedema, erythromyalgia, porphyria, lichen sclerosis, graft-versus-host disease, and diabetic cheiroarthropathy, were also considered in the differential diagnosis and excluded if recognized. Further exclusion criteria were HBV and/or HCV infection, alcoholism, autoimmune liver diseases (i.e., primary biliary cholangitis, autoimmune hepatitis), liver cirrhosis, malignancy, cardiac disease, or dialysis. All patients provided written informed consent for data storage and analysis. The study was carried out in accordance with the Declaration of Helsinki and its later amendments and was approved by the ethics committee of the teaching hospital of the “Luigi Vanvitelli” University of Campania, Naples, Italy.

### 2.2. Clinical Features and Visceral Involvement

All patients underwent a physical examination and treatment for SSc as appropriate. The following parameters were recorded: height, weight, body mass index (BMI), extent of cutaneous involvement, presence of telangiectasias, calcinosis, digital ulcers, pitting scars, Raynaud’s phenomenon, arthritis, esophageal involvement (X-ray hypomotility, with or without dysphagia), and scleroderma renal crisis (proteinuria > 300 mg/24 h). Pulmonary involvement, with a particular focus on fibrosis, was also carefully investigated by means of high-resolution computed tomography (HRCT), spirometry (to search for a restrictive pulmonary function test pattern), and diffusing lung capacity for carbon monoxide (DLCO). M-mode, two-dimensional (2D), and color Doppler flow conventional echocardiography was also carried out in all patients to evaluate pulmonary systolic hypertension (PAPs) and diastolic dysfunction through transmitral E/A ratio (E/A) assessment. Fifty-four of the 59 patients underwent nailfold videocapillaroscopy (NVC), and the following patterns were used for classification [12]: (1) early NVC pattern (E), which included few capillary haemorrhages and enlarged/giant capillaries, a relatively well-preserved capillary distribution, and no evident loss of capillaries; (2) active NVC pattern (A), characterized by frequent giant capillaries and capillary haemorrhages, a moderate loss of capillaries, a mildly disorganized capillary architecture, some avascular areas, and absent/mild ramified capillaries; and (3) late NVC pattern (L), typically represented by irregular enlargement of the capillaries, few/absent giant capillaries, no evidence of haemorrhages, severe loss of capillaries and large avascular areas, a harshly disorganized normal capillary array, and frequent ramified/bushy capillaries [12]. SSc disease activity was assessed according to the criteria proposed by Valentini et al. [13], which allow disease activity assessment in the global SSc population as well as in the two main subsets, namely, diffuse and limited SSc [3].

### 2.3. Serological Parameters

The following serological markers were evaluated using established techniques: anti-nuclear antibodies (ANAs, by indirect immunofluorescence on the Hep-2 cell line, with results expressed in titers and patterns); anti-extractable nuclear antigens (ENAs), including anti-topoisomerase (anti-Scl-70), anti-centromere (ACA), and anti-RNA polymerase III (RNA Pol III); complement C3 and C4; inflammatory markers, such as C-reactive protein (CRP) and erythrocyte sedimentation rate (ESR); total cholesterol; LDL and HDL cholesterol; triglycerides; and vitamin D. Briefly, ANAs and ENAs were used to categorize SSc patients into diffuse or limited subsets [1]; C3, C4, CRP, and ESR served as serological disease activity markers; serum lipoproteins were assayed in order to make correlations with CAP findings. All patient sera were internally processed at the laboratories of the “Luigi Vanvitelli” University of Campania teaching hospital, which are subjected to periodic quality control assessments, thus ensuring the reliability of results and comparisons.

### 2.4. Procedures

Anthropometric clinical data and blood samples were collected on the same date as the TE examination. The FibroScan^®^ Mini+ 430 powered by CAP (Echosens SA, Créteil, France) was used for the present study (Figure 1). FibroScan^®^, powered with CAP (i.e., controlled attenuation parameter), is a non-invasive and inexpensive diagnostic tool able to evaluate liver fibrosis by stiffness (rigidity) measurement, expressed in kilopascal (kPa), and is able to quantify hepatic steatosis with good accuracy [14,15,16,17]. The TE technique and procedure have improved over time [8,9]. At the beginning, it relied on a transient mechanical vibration, which was used to create a distortion in the tissue (share wave) by vibrating the skin through a motor. Young’s modulus, deduced under the hypothesis of homogeneity, isotropy, and pure elasticity (E = 3ρV^2^), reproduced an image of the motion of that distortion as it passed deeper into the body using a 1D ultrasound beam and provided a quantitative one-dimensional (i.e., a line) image of “tissue” stiffness. As this technique improved, a specific implementation of 1D transient elastography, called vibration-controlled transient elastography, was developed to assess average liver stiffness, which correlated with liver fibrosis assessed by liver biopsy. Finally, a further implementation led to the possibility of evaluating the CAP, which was first validated as an estimate of ultrasonic attenuation at 3.5 MHz using Field II simulations and tissue-mimicking phantoms [8,9].

The quality of TE is based on the congruence of ten measurements [18]. In particular, an interquartile range of <30% represents a reliable test threshold according to the manufacturer’s instructions. TE is a reproducible test; however, there are some elements to consider when carrying out the exam. Some points of the liver parenchyma, especially those closer to the capsule or vascular structures, have an altered LS; moreover, failure to comply with fasting can also lead to an overestimation of LS. In this case, the TE value may not be correct despite a low IQR [18]. In addition, the different physical conformation of the rib cage, the presence of excessive fat, and the different hepatic volumes are elements that may affect the correct identification of the suitable intercostal space and the distance from the anterior or middle axillary line as points on which to lay the probe for the ten measurements [18,19]. In this regard, the use of the ultrasound probe to identify an optimal measurement point is a modality that can offer advantages [20]. The latest generation of FibroScan^®^ instruments have a standard ultrasound probe on board that allows visualization of the liver segment on which to perform the measurement [21]. An additional parameter to consider for maintaining the quality of the examination is the elastogram pattern that appears on the screen for each measurement. Three categories have been described based on the length of the graphic representation and shear wave dispersal (level of parallelism displayed in the elastogram) [22]. A comparison with biopsy data showed that the diagnostic accuracy of TE was significantly greater if based on the quality of each individual measurement.

Another important element that can affect the quality of the examination is the subcutaneous fat related to the condition of overweight/obesity. In this case, an excessive thickness of subcutaneous fat increases the distance between the skin surface and the liver capsule. In such circumstances, clinical studies have shown that the use of the XL probe improves the accuracy of LS in difficult patients with a body mass index (BMI) > 28 [23,24]. In light of these data, training in the use of FibroScan should always include knowledge of all cofactors that can impact performance. Interoperator variability and low IQR/M should be taken into account along with the other elements described above.

In this study, CAP and liver stiffness were measured by the same experienced operator aware of all the issues and pitfalls discussed above. An M probe (3.5 MHz) was used, which was placed on the skin in the intercostal space over the right lobe of the liver (Figure 2). The XL probe was not needed, as the median BMI in our patient population was 24.77 (Table 1). Patient preparation involved a 6 h fast preceding the examination (drinks were allowed, but only non-carbonated beverages were permitted). As per good practice, liver stiffness was calculated over at least 10 valid measurements, with a ratio of the interquartile range (IQR) to the median of the liver stiffness (IQR/Median) of ≤25% [25,26,27]. Notably, CAP, through the SmartExam program, was continuously computed during the entire examination until the CAP gauge reached 100%. The hepatic steatosis grade was defined by CAP cut-off values previously established. A CAP value < 238 dB/m denoted the absence of steatosis (S0). CAP values ≥ 238 ≤ 259 dB/m denoted mild steatosis (grade S1), CAP values ≥ 260 ≤ 290 dB/m denoted moderate steatosis (grade S2), and CAP values ≥291 dB/m denoted severe steatosis (grade S3). The cut-off value for severe hepatic fibrosis was set at >7.0 kPa [28,29,30,31,32]. Patients were classified as having no fibrosis (F0, LS < 5.2 kPa), mild to moderate fibrosis (F1–2, LS between 5.2 and 7 kPa), and severe fibrosis (F3, LS values > 7 kPa) (Figure 3).

### 2.5. Statistical Analysis

Anthropometric, clinical, biochemical, and instrumental data were registered in a database and analyzed with MedCalc v.18.10.2 (MedCalc Software, Mariakerke, Belgium). For continuous variables, the measures of centrality and dispersion were medians and interquartile ranges (IQR). The Spearman’s rank correlation rho test was used to analyze the association between categorical variables. All variables found to have at least a trend (*p* value ≤ 0.10), suggesting an association with liver fibrosis or liver steatosis via univariate analysis, were entered into a multivariate logistic regression model with a stepwise approach. The threshold for statistical significance was set at *p* < 0.05 (two-tailed).

## 3. Results

Clinical and demographic features of the 59 patients recruited in the study are detailed in Table 1. The median age of patients (89.8% women) was 51 years (range 20–80), and the median disease duration was 6 years (range 1–17). Clinical subsets of disease were as follows: 27.1% of patients had a sine-scleroderma phenotype, 59.3% were classified as having limited SSc, and diffuse SSc was observed in 13.5% of patients.

Overall, 55.9% of patients had gastrointestinal involvement, 35% had interstitial lung disease, and 1.7% were diagnosed with pulmonary hypertension; only 1.7% of patients displayed digital ulcers. With regard to SSc treatment, 54.2% of patients were on immunosuppressants (43.7% on azathioprine and 56.3% on mycophenolate), 20.3% were receiving hydroxychloroquine, and 64.4% were taking steroids (prednisone doses ≥ 5 mg and ≤10 mg) at the time of the evaluation.

According to the capillaroscopic pattern [12], 44.4% of patients showed an early pattern, 27.8% had an active pattern, 11.1% presented with a late pattern, whereas 16.7% displayed nonspecific alterations.

Autoantibody prevalence was as follows: 33.9% of patients tested positive for anti-centromere antibodies, 40.7% had anti-Scl-70 antibodies, 8.5% were positive for RNA polymerase III autoantibodies, and 16.9% did not show any conventional autoantibodies.

Concerning liver fibrosis, a median LS of 4.5 (2.9–8.3) kPa was recorded. Fibrosis was not detected (F0) in 69.5% of patients; 27.1% showed LS values between 5.2 and 7; and 3.4% displayed LS values > 7 (F3), suggesting significant liver fibrosis. For liver steatosis, the median CAP value was 223 dB/m (IQR: 164–343). In detail, 66.1% of patients showed CAP values <238 consistent with no steatosis; 15.2% had CAP values ≥ 238 ≤ 259 reflecting mild (S1) steatosis; 13.5% had moderate (S2) steatosis, as suggested by CAP values ≥ 260 to ≤290; and 5.1% were deemed to harbor severe steatosis (S3) due to CAP values ≥ 291.

Table 2 shows the univariate analysis results according to the LS and CAP findings. Briefly, gender (*p* = 0.013), HDL-cholesterol (*p* = 0.014), triglycerides (*p* = 0.006), and telangiectasias (*p* = 0.045) were all significantly associated with LS values, while activity index (*p* = 0.007), PAPs (*p* = 0.023), E/A (*p* = 0.002), BMI (*p* < 0.0001), age (*p* < 0.0001), and MMF (*p* = 0.013) were all significantly associated with CAP values.

According to multiple regression analysis by the stepwise method (Table 3), only high triglyceride levels were significantly correlated with liver fibrosis in SSc patients (*p* = 0.02); conversely, BMI and age were shown to be significantly correlated with liver steatosis (*p* = 0.023 and *p* = 0.022, respectively).

## 4. Discussion

Fibrosis is the pathological hallmark underlying much of the morbidity and mortality associated with SSc and should therefore be regarded as a lethal component of the disease [33]. The other two key features of SSc are immunological abnormalities and vasculopathy. The interplay between these factors likely results in the pathological changes seen in SSc [33,34]. However, sorting out the role of each mechanism in determining SSc pathogenesis has thus far been a daunting task. Indeed, several experimental observations have suggested immune dysregulation as a cause of, or at least a contributor to, fibrosis in SSc; on the other hand, fibrosis has also been shown to contribute to aberrant immune cell activation [33,34]. In addition, both immune abnormalities and fibrosis are also linked to vasculopathy in SSc, with vascular damage also shown as an activator of immune cells in experimental models. Whatever the mechanism, persistent fibroblast activation and increased myofibroblast differentiation lead to excessive extracellular matrix deposition and, in turn, distortion of tissue architecture, impairment of organ function, and, eventually, organ failure [33,34]. Fibrosis in SSc occurs mainly in the skin but may progress to visceral organs, particularly the heart, the lungs, and the gastrointestinal system [3]. With regard to the latter, the gastrointestinal tract is indeed the most commonly affected internal organ in SSc since up to 90% of patients experience symptoms related to upper and/or lower gastrointestinal dysmotility, which may be associated with significant morbidity and mortality [35]. Specifically, esophageal dysmotility is often one of the earliest features of SSc and may present with symptoms of dysphagia, heartburn, and regurgitation; gastric involvement may present with symptoms of gastroparesis (i.e., early satiety, bloating, and regurgitation); while small intestine involvement may be suggested by symptoms of small intestinal dysmotility (e.g., distention, bloating), small intestinal bacterial overgrowth, or both. Finally, constipation and fecal incontinence may suggest colon and anorectal involvement [35]. Conversely, distortion of the hepatic architecture by fibrosis in SSc patients has not been adequately investigated. This may be explained by the fact that autoimmune hepatobiliary conditions represent the best-known liver comorbidities affecting SSc patients [6,7] and because of the need for invasive procedures (i.e., liver biopsy) to confirm the diagnosis. The availability of transient elastography has now changed this scenario, as it allows for easy, non-invasive measurement of liver fibrosis as well as steatosis, with results that have been shown to agree with pathological examination of liver tissue following hepatic biopsy [22]. This study therefore aimed at assessing the prevalence and predictors of significant liver fibrosis and steatosis in SSc patients by means of TE. We speculated that liver fibrosis may possibly be found in SSc patients, even if asymptomatic or in the absence of liver biochemistry test alterations. The reasons behind our hypothesis lie in the widespread involvement of organs and tissues in SSc patients as well as in the assumption that some of the known organ fibrotic changes may be asymptomatic for years; for instance, lung fibrosis may be detected through HRCT and/or DLCO even in asymptomatic patients [3]. Finally, it may also be hypothesized that fibrosis in the liver may not become symptomatic because it progresses slower than in other organs or because patients may die of other complications before liver fibrotic changes become clinically apparent. Indeed, consistent with our hypothesis, most patients in our cohort displayed only moderate liver fibrosis. Specifically, we found that the prevalence of significant liver fibrosis (F3) was relatively low (3.4%), but moderate liver fibrosis (F1–F2) was relatively high (27.1%). Our data are consistent with previously published data, reporting a prevalence of 1–9% for significant liver fibrosis in SSc patients [6,36,37]. The prevalence of significant liver fibrosis may appear to be low, considering that SSc is characterized by fibrosis involving several tissues and organs. Again, as hypothesized above, the degree of fibrosis may nonetheless depend on the different rates of progression in different tissues. It should also be remembered, however, that patients with viral hepatitis, drug or alcohol abuse, or other causes of significant liver disease, including autoimmune hepatobiliary conditions, were excluded from the study.

In real clinical settings, when SSc patients show no evidence of liver disease or abnormalities in liver laboratory tests, the possibility of fibrosis is typically ignored. Our study, however, suggests that moderate fibrosis may still be detected even if clinically silent.

In this study, the association between LS and CAP values and the epidemiological and clinical features of SSc patients was also investigated. Regarding LS, only high serum triglyceride levels correlated with liver fibrosis. This is not surprising since this factor is a known predictor of fibrosis development in the general population as well, as lipids are mainly stored as triglycerides, an inert and non-cytotoxic form of lipids, in the liver [38].

We also evaluated the possible effects of concurrent immunosuppressive drug therapy on liver fibrosis. However, we did not find any relevant effects of these drugs on liver fibrosis or steatosis risk, presumably due to the limited number of patients in our cohort.

With regard to liver steatosis, the estimated prevalence of severe steatosis, as per CAP values above 291 dB/m, was around 5.1% and did not differ from that of the general population [30]. Moderate (S2) steatosis was recorded in 13.5% of patients. The predictors of liver steatosis were found to be the same as those involved in otherwise healthy subjects [32]. Interestingly, although an elevated BMI was recognized as a predictor of liver steatosis, the median BMI in our SSc population was 24.77 (18.93–36.51), similar to that of the general Italian population, as recently reported by Maffoni et al. (22.5, IQR: 20.3–25.2) [39]. According to multiple regression analysis with the stepwise method, only BMI and age were significantly correlated with liver steatosis. Again, we also evaluated the possible effects of concurrent immunosuppressive drug therapy on liver steatosis risk. Although linear regression identified the use of MMF as a possible risk factor for steatosis, multiple regression analysis did not confirm this association.

In conclusion, we reported a low prevalence of marked fibrosis (3.4% of patients), which is the same as that expected in the general population, in our SSc patient population. Moderate fibrosis, affecting a larger proportion of our patients, may nonetheless be a SSc “signature”, albeit asymptomatic. Likewise, the prevalence of significant steatosis was low and related to the same variables associated with steatosis in the general population.

In the medical literature, only a few studies have investigated the prevalence of liver fibrosis in the general population, and an accurate LS cut-off value in this population has not yet been established; therefore, prevalence estimates vary depending on the chosen LS value. Koehler et al. [40] reported a 5.6% prevalence of liver fibrosis, with a cut-off value of 8.0 kPa, in 3040 subjects older than 45 years in Rotterdam. In a study from Hong Kong, the estimated prevalence among 922 subjects aged 18 to 72 years was 2%, with a cut-off of 9.6 kPa [41]. Finally, in a study from France, including 1358 subjects older than 45 years, the estimated prevalence was 7%, with a predefined cut-off value of 8 kPa [42]. In all three studies, the most common cause of liver disease was non-alcoholic fatty liver disease (NAFLD). Although we found a low prevalence of marked fibrosis, our study suggests that TE is a valuable method for detecting significant liver fibrosis in subjects with no known liver disease and is useful for screening for liver fibrosis in SSc patients. Finally, since liver steatosis in our SSc patients appeared to be favored by the same factors that increase the risk in the general population, the same preventive and therapeutic measures for fatty liver disease should be advised [43].

There are, of course, some limitations to our study that need to be acknowledged. The first one is the small sample size of our patient population, although the rarity of SSc may be at least a partial justification for this issue; multicenter studies would be needed to sum up significant numbers of patients. In addition, the cross-sectional design did not allow us to draw conclusions about the variations in liver fibrosis over time. 

## 5. Conclusions

Based on the findings of the current study, we can cautiously conclude that SSc patients do not appear to be at risk of significant liver fibrosis, despite the fact that fibrotic changes are usually detected in several other organs and tissues in patients affected by this disease, and moderate fibrosis may still be found in the liver. Likewise, liver steatosis in SSc patients appears to be a consequence of the same predisposing factors acting on the general population. Therefore, SSc patients should abide by the same preventive and therapeutic measures suggested for the reduction of the impact of NAFLD in the general population [43]. Finally, TE has been shown to be a valuable diagnostic procedure for screening and detection of liver fibrosis in SSc patients with no known concomitant liver disease due to different etiologies (viral, alcoholic, etc.). The limitations of this study warrant further investigation in larger series of patients with longitudinal evaluation of fibrosis outcome.

## Figures and Tables

**Figure 1 diagnostics-13-01766-f001:**
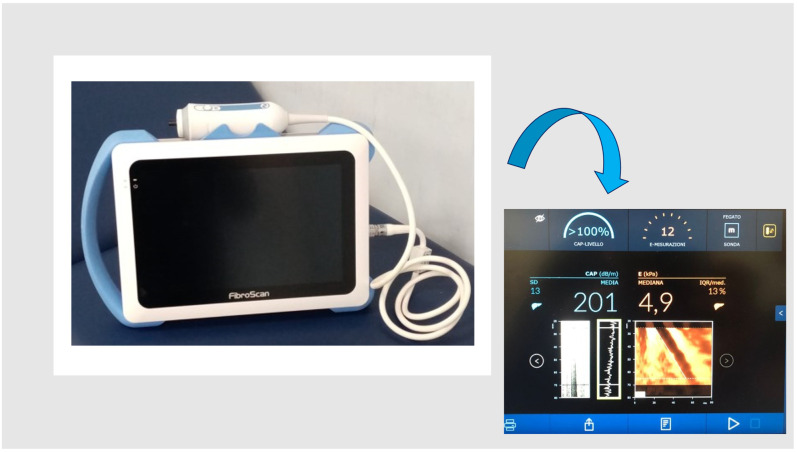
FibroScan^®^ Mini+ 430 (Echosens SA, Créteil, France).

**Figure 2 diagnostics-13-01766-f002:**
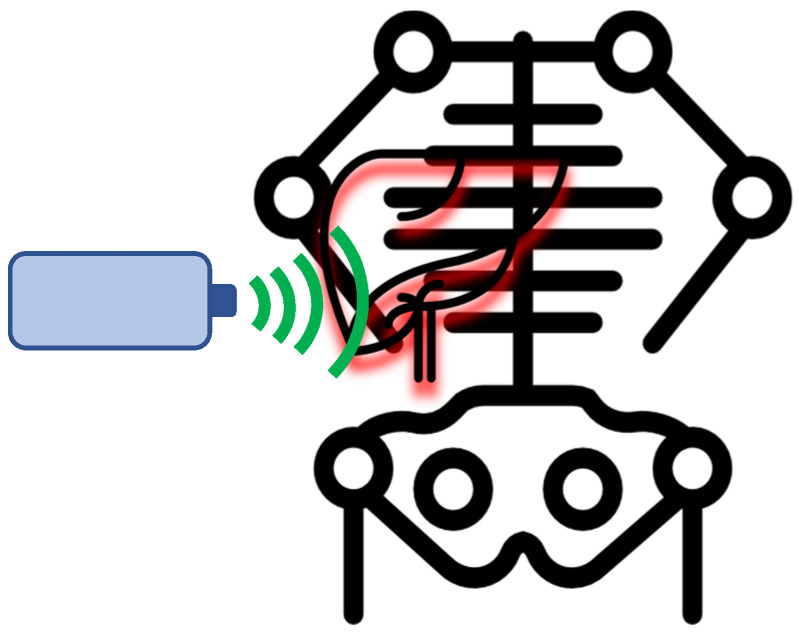
Schematic illustration of the non-invasive assessment of liver fibrosis and steatosis by transient elastography using the FibroScan^®^ procedure.

**Figure 3 diagnostics-13-01766-f003:**
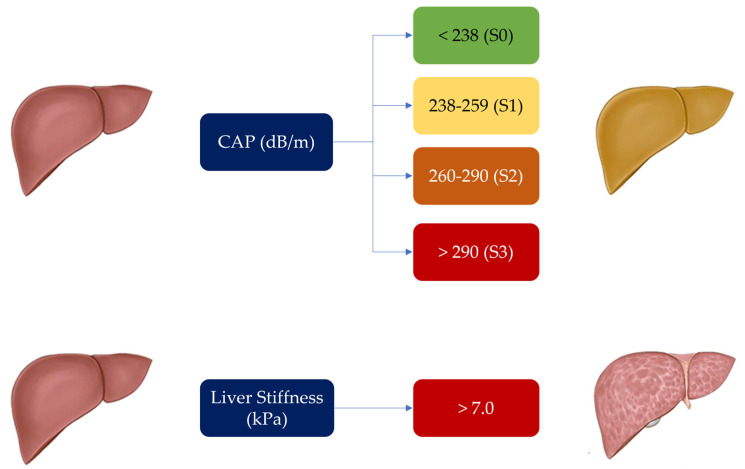
Controlled Attenuation Parameter (CAP) and liver stiffness cut-offs used for determination and degree of liver steatosis and fibrosis in our patient population.

**Table 1 diagnostics-13-01766-t001:** Main characteristics of the study population (n = 59).

*Clinical and epidemiological data*	
Age (years), median (IQR)	51 (20–80)
Sex, n (%) M/F	6 (10.2%)/53 (89.8%)
BMI (kg/m^2^), median (IQR)	24.77 (18.93–36.51)
*SSc characteristics*	
Duration of disease in years, median (IQR)	6 (1–17)
Activity index, median (IQR)	0.5 (0–4)
mRSS, mean (SD)	2.24 (2.95)
Cutaneous involvement, subset, n (%)	
Sine	16 (27.1%)
Limited	35 (59.3%)
Diffuse	8 (13.5%)
E/A ratio, n (%) normal/abnormal (54)	36 (66.6%)/18 (33.3%)
PAPs, median (IQR)	25 (0–65)
FVC, median (IQR)	98.5 (57.4–126)
DLCO, median (IQR)	85.5 (38–138)
Videocapillaroscopic pattern, n (%)	54 (91.5%)
Normal	9 (16.7%)
Early	24 (44.4%)
Active	15 (27.8%)
Late	6 (11.1%)
Gastrointestinal involvement, n (%)	33 (55.9%)
Scleroderma renal crisis, n (%)	-
ILD, n (%), 57 pts	20 (35%)
ANA positive, n (%)	59 (100%)
ENA, n (%)	59 (100%)
no autoantibodies	10 (16.9%)
anti-centromere	20 (33.9%)
anti-Scl-70	24 (40.7%)
anti-RNA polymerase III	5 (8.5%)
Ulcers, n (%)	1 (1.7%)
Pitting scars, n (%)	7 (11.9%)
Telangiectasias, n (%)	28 (48%)
*FibroScan results*	
CAP median (IQR)	223 (164–343)
LS median (IQR)	4.5 (2.9–8.3)
*Concomitant therapies*	
Corticosteroids, n (%)	38 (64.4%)
Hydroxychloroquine, n (%)	12 (20.3%)
Immunosoppressants, n (%)	32 (54.2%)
Azatioprine, n (%)	14 (23.7%)
Micofenolate, n (%)	18 (30.5%)
*Laboratory parameters*	
Total cholesterol, mg/dL, median (IQR), 52 pts	182.5 (101–307)
HDL-cholesterol, mg/dL, median (IQR), 40 pts	64.5 (37–130)
LDL-cholesterol, mg/dL, median (IQR), 36 pts	102.5 (40–172)
Triglycerides, mg/dL, median (IQR), 46 pts	87 (33–268)
Vitamin D, UI, median (IQR), 47 pts	28.5 (7.7–53.9)

Abbreviations: ANA = Anti-Nuclear Antibodies; BMI = Body Mass Index; CAP = Controlled Attenuation Parameter; DLCO = Diffusing Capacity of the Lungs for Carbon Monoxide; ENA = Extractable Nuclear Antigens; F = Female; FVC = Forced Vital Capacity; HDL = High Density Lipoprotein; ILD = Interstitial Lung Disease; IQR = Interquartile Range; LDL = Low Density Lipoprotein; LS = Liver Stiffness; M = Male; mRSS = modified Rodnan Skin Score; n = number; PAPs = Systolic Pulmonary Artery Pressure; SD = Standard Deviation; and SSc = Systemic Sclerosis.

**Table 2 diagnostics-13-01766-t002:** Linear regression analysis between liver fibrosis and liver stiffness values and other clinical variables (patients, n = 59).

Parameter						
	LS			CAP		
	Correlation Coefficient	95% CI	*p*	rho	95% CI	*p*
Subset sine/L/D	0.24	−0.02 to 0.47	0.068	0.06	−0.19 to 0.32	0.6
L/D	0.24	−0.01 to 0.48	0.059	0.02	−0.24 to 0.28	0.9
Gender	0.32	0.06 to 0.54	**0.013**	0.17	−0.10 to 0.41	0.2
HDL-cholesterol	−0.38	−0.63 to −0.07	**0.014**	0.11	−0.21 to 0.42	0.4
TG	0.40	0.11 to 0.62	**0.006**	0.27	−0.03 to 0.53	0.06
ILD	0.23	−0.04 to 0.46	0.09	0.23	−0.04 to 0.47	0.09
Telangiectasias	0.26	−0.001 to 0.49	**0.045**	−0.09	−0.35 to 0.18	0.5
DLCO	−0.23	−0.46 to 0.04	0.087	−0.12	−0.37 to 0.15	0.4
Activity index	0.16	−0.11 to 0.40	0.2	0.34	0.09 to 0.56	**0.** **007**
PAPs	−0.08	−0.35 to 0.20	0.5	0.31	0.04 to 0.54	**0.023**
E/A	0.14	−0.14 to 0.40	0.3	0.41	0.15 to 0.62	**0.002**
BMI	0.09	−0.18 to 0.34	0.5	0.50	0.27 to 0.67	**<0.0001**
Age	−0.17	−0.42 to 0.09	0.2	0.52	0.29 to 0.69	**<0.0001**
Immunosuppressive treatment	0.14	−0.12 to 0.38	0.3	0.24	−0.01 to 0.47	0.06
MMF	0.16	−0.10 to 0.41	0.22	0.32	0.07 to 0.53	**0.013**
AZA	0.05	−0.21 to 0.3	0.7	−0.08	−0.34 to 0.17	0.5

Abbreviations: L = limited; D = diffuse. AZA = Azatioprine; BMI = Body Mass Index; CAP = Controlled Attenuation Parameter; CI = Confidence Interval; DLCO = Diffusing Capacity of the Lungs for Carbon Monoxide; HDL = High Density Lipoprotein; ILD = Interstitial Lung Disease; LS = Liver Stiffness; MMF = Mycophenolate Mofetil; PAPs = Systolic Pulmonary Artery Pressure; and TG = Triglycerides. Bold type of *p* values identifies significant results.

**Table 3 diagnostics-13-01766-t003:** Multiple regression analysis according to LS and CAP (n = 59).

Parameter								
	LS				CAP			
	Coefficient	Std. Error	t	*p*	Coefficient	Std. Error	t	*p*
Gender	0.43	0.82	0.53	0.6				
HDL-cholesterol	−0.01	0.01	−1.23	0.23				
Triglycerides	0.01	0.004	2.43	**0.02**				
Telangectasias	0.32	0.34	0.95	0.35				
Activity index					4.61	4.12	1.12	0.26
PAPs					0.33	0.31	1.08	0.28
E/A					11.7	10.17	1.12	0.25
BMI					2.28	0.96	2.36	**0.023**
Age					0.77	0.33	2.37	**0.022**
MMF					9.72	8.35	1.16	0.25

Abbreviations: BMI = Body Mass Index; CAP = Controlled Attenuation Parameter; HDL = High Density Lipoprotein; LS = Liver Stiffness; MMF = Mycophenolate Mofetil; and PAPs = Systolic Pulmonary Artery Pressure. Bold type of *p* values identifies significant results.

## Data Availability

Raw data are available from the corresponding author upon reasonable request.

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
