# Peer review of "Liver Involvement in Patients with Systemic Sclerosis: Role of Transient Elastography in the Assessment of Hepatic Fibrosis and Steatosis"

_diagnostics, 2023, doi:10.3390/diagnostics13101766_

Round 1
Reviewer 1 Report
In the introduction section, the authors cited many references, but only a couple of them are recent ones. I advise the authors to revise the introduction section with the latest and more prominent references.
The background of the study should be clarified and mentioned at the end of the abstract. The importance of the study should be justified.
Why the role of transient elastography in assessing hepatic fibrosis and steatosis has been taken into count should be adequately explained.
If possible, the workflow should be explained as a proper flow chart for the readers to understand.
The numerical and statistical values should be carefully rechecked to avoid any mistakes.
The serological parameters section should be elaborated more.
The entire manuscript should be revised, and the grammatical errors should be rectified. A proper English language needs to be done.
The authors should add their perspectives on the results and clearly state the study's limitations.
The authors are advised to improve the quality of the figures if possible.
Extensive editing of the English language is required
Reviewer 2 Report
Abstract: Please indicate only percentages rather than total numbers.
Write the prevalence of systemic sclerosis in Italian Population
On what basis, 59 samples was calculated?
Can author show the sample size calculation?
Write the inclusion and exclusion criteria of the subejcts involved in this study
In section 2.2, please cite the following reference for BMI (https://www.ncbi.nlm.nih.gov/pmc/articles/PMC8017326/)
The section 2.2 was very clear.
The results are fine but authors need to remove the total numbers and add only the percentages
Discussion was found not to be strong and authors need to discuss what is the importance of this study.
Based on small sample size, author may not conclude and it can be recommended as this study has no risk of significant liver fibrosis and so on.
Authors need to mention as small sample size was one of the limitations of this study
Reviewer 3 Report
The study is interesting and aimed at evaluating the presence of a rare disease manifestation in a cohort of patients with systemic sclerosis.
It would be interesting to know:
- Presence of other autoimmune diseases, particularly primary biliary cirrhosis (in the materials and methods section it is described that only patients with autoimmune hepatitis and not PBC were excluded)
- Frequency of positivity for AMA
- Although they do not always correlate with liver damage, it would be interesting to know whether patients with liver fibrosis had different transaminase and GGT values than other patients.
The same information would also be useful for the group of patients with hepatic steatosis.
- Since only half of the patients (54.2) were taking an immunosuppressant, it is safe to assume that the patients who were not taking these drugs had mild or early-stage disease. In fact, only 6 patients presented with a "late" capillaroscopic picture and only 15 with an "active" picture. What is the prevalence of liver fibrosis in patients with more severe disease and/or long history of disease?
- It would also be interesting to know whether the presence of liver fibrosis is associated with positivity of any autoantibody or gastrointestinal involvement
- 64% of patients were taking glucorticoids. What is the median dosage of the cohort?
It is reported in the materials and methods that serum levels of C3 and C4 were also assessed. They are not reported in the table.
Minor editing of English language required
Round 2
Reviewer 1 Report
The revised version of the manuscript looks fine and can be recommended for publication.
Reviewer 2 Report
Pease add the relevant reference for BMI (https://www.tandfonline.com/doi/full/10.2147/DMSO.S294948)
Reviewer 3 Report
The authors addressed all the major issues.
The paper can now be published
please read again to avoid some missclick mistakes